



# Influence of temporally varying weatherability on CO$_2$–climate coupling and ecosystem change in the late Paleozoic.

Jon D. Richey[1*], Isabel P. Montañez[1*], Yves Goddéris[2], Cindy V. Looy[3], Neil P. Griffis[1,4], William A. DiMichele[5]

[1]Department of Earth and Planetary Sciences, University of California, Davis, Davis, CA 95616, USA.
[2]Géosciences Environnement Toulouse, CNRS – Université Paul Sabatier, Toulouse, France.
[3]Department of Integrative Biology and Museum of Paleontology, University of California, Berkeley, Berkeley, CA 94720, USA.
[4]Berkeley Geochronology Center, Berkeley, CA 94720, USA.
[5]Department of Paleobiology, Smithsonian Museum of Natural History, Washington, DC 20560, USA.

[*]*Correspondence to*: Jon D. Richey (jdrichey@ucdavis.edu); Isabel P. Montañez (ipmontanez@ucdavis.edu)

**Abstract** Earth's penultimate icehouse, the Late Paleozoic Ice Age (LPIA), was a time of dynamic glaciation and repeated ecosystem perturbation, under conditions of substantial variability in atmospheric $p$CO$_2$ and O$_2$. Improved constraints on the evolution of atmospheric $p$CO$_2$ and O$_2$:CO$_2$ during the LPIA and its subsequent demise to permanent greenhouse conditions is crucial for better understanding the nature of linkages between atmospheric composition, climate, and ecosystem perturbation during this time. We present a new and age-recalibrated $p$CO$_2$ reconstruction for a 40-Myr interval (~313 to 273 Ma) of the late Paleozoic that (1) confirms a previously hypothesized strong CO$_2$-glaciation linkage, (2) documents synchroneity between major $p$CO$_2$ and O$_2$:CO$_2$ changes and compositional turnovers in terrestrial and marine ecosystems, (3) lends support for a modeled progressive decrease in the CO$_2$ threshold for initiation of continental ice sheets during the LPIA, and (4) indicates a likely role of CO$_2$ and O$_2$:CO$_2$ thresholds in floral ecologic turnovers. Modeling of the relative role of CO$_2$ sinks and sources, active during the LPIA and its demise, on steady-state $p$CO$_2$ using an intermediate complexity climate-C cycle model (GEOCLIM) and comparison to the new multi-proxy CO$_2$ record provides new insight into the relative influences of the uplift of the Central Pangaean Mountains, intensifying aridification, and increasing mafic rock to-granite rock ratio of outcropping rocks on the global efficiency of CO$_2$ consumption and secular change in steady-state $p$CO$_2$ through the late Paleozoic.



## 1 Introduction

Earth's penultimate and longest-lived icehouse (340 to 290 Ma) occurred under the lowest atmospheric $CO_2$ concentrations of the last half-billion years (Foster et al., 2017) and, potentially, the highest atmospheric $pO_2$ of the Phanerozoic (Glasspool et al., 2015; Krause et al., 2018; Lenton et al., 2018). Anomalous atmospheric composition, along with 3% lower solar luminosity (Crowley and Baum, 1992), may have primed the planet for a near-miss global glaciation (Feulner, 2017). Notably, Earth's earliest tropical forests assembled and expanded during this icehouse (the Late Paleozoic Ice Age; LPIA), leading to the emergence of large-scale wildfire. Paleotropical terrestrial ecosystems underwent repeated turnovers in composition and architecture, culminating in the collapse of wetland (coal) forests throughout tropical Pangea at the close of the Carboniferous (Cleal and Thomas, 2005; DiMichele, 2104), possibly promoting the diversification and ultimate dominance of amniotes (Pardo et al., 2019). In the marine realm, global rates of macroevolution (origination, extinction) decreased, in particular among tropical marine invertebrates, and genera with narrow latitudinal ranges went extinct at the onset of the LPIA (Stanley, 2016; Balseiro and Powell, 2019). Low marine macroevolutionary rates continued through to the demise of the LPIA in the early Permian (Stanley and Powell, 2003; McGhee, 2018).

Reconstructions of late Paleozoic atmospheric $pCO_2$ document a broad synchroneity between shifts in $CO_2$, glaciation history, glacioeustasy, and restructuring of paleotropical biomes, underpinning the hypothesized greenhouse-gas forcing of sub-million-year glacial-interglacial cycles (Montañez et al., 2016) and the terminal demise of the LPIA (Montañez et al., 2007). For late Paleozoic $pCO_2$ (and $pO_2$) reconstructions, however, broad intervals of low temporal resolution and significant uncertainties limit the degree to which mechanistic linkages between atmospheric composition, climate, and ecosystem change can be further evaluated. Moreover, the potential impact of large magnitude fluctuations in atmospheric $O_2$:$CO_2$, which characterized the late Paleozoic, on the biosphere has been minimally addressed. On longer timescales ($\geq 10^6$ yr), the relative role of potential $CO_2$ sinks and sources on secular changes in late Paleozoic atmospheric $CO_2$ and, in turn, as drivers of glaciation and its demise, remain debated (McKenzie et al., 2016; Goddéris et al., 2017; Macdonald et al., 2019).

Here, we present a multi-proxy atmospheric $pCO_2$ reconstruction for a 40-Myr interval (313 to 273 Ma) of the late Paleozoic, developed using new leaf fossil-based estimates integrated with recently published and age-recalibrated Pennsylvanian $pCO_2$ estimates of $10^5$-yr resolution (Montañez et al., 2016), and re-evaluated fossil soil- (paleosol) based



$CO_2$ estimates for the early Permian (Montañez et al., 2007). Our new multi-proxy record offers higher temporal resolution
than existing archives while minimizing and integrating both temporal and $CO_2$ uncertainties. This $pCO_2$ reconstruction,
together with new $O_2:CO_2$ estimates of similar temporal resolution, permits refined interrogation of the potential links
between fluctuations in atmospheric composition, climate shifts, and ecosystem events through Earth's penultimate icehouse.
Moreover, comparison of the new 40-Myr $CO_2$ record with modeled steady-state $pCO_2$ and seawater $^{87}Sr/^{86}Sr$ over the same
interval provides new insight into the relative importance and evolution of $CO_2$ sinks and sources during late Paleozoic
glaciation and its turnover to a permanent greenhouse state.

## 2 Materials and Methods

A brief account of the methods is presented here; more details are presented in the Supplementary Materials and Methods.
Primary data generated or used in this study is deposited in the Dryad Digital Repository (Richey et al., 2020) and can be
accessed at https://doi.org/10.25338/B8S90Q.

### 2.1 Sample Collection and Analysis

To build the $pCO_2$ record, 15 plant cuticle fossil species/morphotypes were used, collected from eight localities in Illinois,
Indiana, Kansas, and Texas, U.S.A., including four well-studied Pennsylvanian interglacial floras (Sub-Minshall [313 Ma;
Šimůnek, (2018)], Kinney Brick [305.7 Ma; DiMichele et al., (2013)], Lake Sarah Limestone [303.7 Ma; Šimůnek, (2018)],
and Hamilton Quarry [302.7 Ma; Hernandez-Castillo et al., (2009a, b, c)]; Figs. 1a, S2–4, Richey et al., (2020)). The
Pennsylvanian estimates were integrated into a previously published $pCO_2$ reconstruction (313 to 296 Ma; Montañez et al.,
(2016)) of $10^5$-yr resolution built using pedogenic carbonates and wet-adapted seed fern fossils (Figs. 2b, S1b). The Permian
estimates were integrated with previously published latest Carboniferous and early Permian pedogenic carbonate-based $CO_2$
estimates (Montañez et al., 2007), derived from paleosols from successions throughout Arizona, New Mexico, Oklahoma,
Texas, and Utah, U.S.A. (Fig. 1a, Richey et al., (2020)). The pedogenic carbonates and leaf fossil cuticles span a broad
region of Pennsylvanian and early Permian tropical Euramerica (Figs. 1b). Ages of samples used in Montañez et al., (2007)
and (2016) were recalibrated and assigned uncertainties using the latest geologic timescale (Ogg et al., 2016) and





biostratigraphic and geochronologic controls (see Supplementary Materials and Methods; Richey et al., (2020)).
Cuticle and organic matter occluded within pedogenic carbonates (OOM) were rinsed or dissolved, respectively, in 3M
HCl to remove carbonates and analyzed at the Stable Isotope Facility, University of California, Davis, using a PDZ Europa
ANCA-GSL elemental analyzer interfaced to a PDZ Europa 20-20 IRMS. External precision, based on repeated analysis of
standards and replicates, is $<\pm0.2$‰. For Hamilton Quarry (HQ), all material was previously mounted on slides for
taxonomic analysis (Hernandez-Castillo et al., 2009a; Hernandez-Castillo et al., 2009b, c). Because of this, biomarker $\delta^{13}C$
values of bulk stratigraphic sediment samples were used (Richey et al., unpublished data; see Supplementary Materials and
Methods). HQ n-$C_{27-31}$ n-alkane $\delta^{13}C$ was analyzed using a Thermo Scientific GC-Isolink connected to a Thermo Scientific
MAT 253. Standard deviation of n-alkane $\delta^{13}C$ was $\pm0.3$‰. For biomarker $\delta^{13}C$, a +4‰ correction was used to account for
fractionation during biosynthesis (Diefendorf et al., 2015) and the standard deviation of all values was used as the
uncertainty (1.6‰, five times the analytical precision).

**2.2 Models**
The MATLAB model Paleosol Barometer Uncertainty Quantification (PBUQ; Breecker, (2013)), which fully propagates
uncertainty in all input parameters, was used to derive pedogenic carbonate-based $CO_2$ estimates (Figs. 2a, S1a). For each
locality, paleosols of inferred different soil orders were modeled separately. We applied improved soil-specific values for
soil-respired $CO_2$ concentrations ($S_{(z)}$; Montañez (2013)) and the $\delta^{13}C$ of organic matter occluded within carbonate nodules
($\delta^{13}C_{OOM}$; Fig. S5) as a proxy of soil-respired $CO_2$ $\delta^{13}C$. For samples where OOM was not recovered, estimates were revised
using PBUQ and the plant fossil organic matter $\delta^{13}C$ used in Montañez et al., (2007) ($\delta^{13}C_{POM}$; Fig. S5). Because of the
limited amount of carbonate nodules remaining after study by Montañez et al., (2007), $\delta^{13}C_{OOM}$ was substituted for $\delta^{13}C_{POM}$
for localities that occur in the same geologic formation and a large error ($\pm 2$‰) was used to account for the uncertainty in
this approach. PBUQ model runs conducted in this study resulted in a small subpopulation of biologically untenable $CO_2$
estimates (i.e., $\leq170$ ppm; Gerhart and Ward, (2010)). To limit estimates below that threshold, two changes to the PBUQ
Matlab code were made (see Supplementary Materials and Methods for details). All other input parameters remained
unchanged from Montañez et al., (2007).



For cuticle fossil-based (Figs. S2–4) $CO_2$ estimates (Fig. 2a, S1a), we utilized a mechanistic (non-taxon-specific) gas-
exchange model (Franks et al., 2014). For some fossil cuticles, pore length (PL) was measured directly; for others, PL was
inferred from guard cell length (GCL; Table S2). Guard cell width was estimated via GCL using the prescribed
gymnosperms and ferns scaler (0.6; Franks et al., (2014); Table S2).
For both stomatal and pedogenic-carbonate-based $CO_2$ modeling, we calculated $\delta^{13}C$ of atmospheric $CO_2$ using the
carbonate $\delta^{13}C$ record generated from an open-water carbonate slope succession (Naqing succession, South China; Buggisch
et al., (2011)), contemporaneous estimates of mean annual temperature (Tabor et al., 2013), and temperature-sensitive
fractionation between low-Mg calcite and atmospheric $CO_2$ (Romanek et al., (1992); Eq. S2; Table S2).
We used the spatially resolved, intermediate complexity GEOCLIM model (Goddéris et al., 2014) to quantitatively
evaluate how steady-state atmospheric $CO_2$ may have responded to changes in weatherability and relative influence of
different $CO_2$ sources and sinks. The spatial distributions of the mean annual runoff and surface temperature were calculated
offline for five time increments (Goddéris et al., 2017) covering the period of interest and for various atmospheric $CO_2$ levels
using the 3D ocean-atmosphere climate model FOAM (Donnadieu et al., 2016). GEOCLIM uses generated lookup tables to
calculate steady-state atmospheric $CO_2$ for a given continental configuration and to account for paleogeography and relief.
Although GEOCLIM model does not include an explicit surface distribution of lithology, weathering rate of mafic rocks and
continental granites are calculated using different methods and the impact of physical erosion on granite weathering is
accounted for (Goddéris et al., 2017). For mafic surfaces, a simple parametric law is used, linking the surface of the
considered grid cell, the local runoff, and mean annual temperature to the local mafic weathering rate. The calibration of the
GEOCLIM model was performed at the continental-scale by tuning the parameters of the model so that 30% of the alkalinity
generated by the weathering of silicates originates from the weathering of mafic rocks (GEOCLIM_REG; Dessert et al.,
2001; Goddéris et al., 2014).

**2.3 $O_2$:$CO_2$**
$O_2$:$CO_2$ ratios (Fig. 3a) were calculated using the 10,000 $CO_2$ estimates produced by our modeling and combined with $O_2$
estimates obtained using geochemical mass balance and biogeochemical models (Krause et al., 2018; Lenton et al., 2018).



Unreasonably high $O_2$:$CO_2$ (generally those that correspond to $CO_2$ ~<200 ppm) were removed from the resulting 10,000
$O_2$:$CO_2$ data set.

**2.4 Statistical Analyses**

We utilize a bootstrap approach that assesses uncertainties of both $CO_2$ (or $O_2$:$CO_2$) and age. Each age uncertainty was
truncated to ensure no overlap in locality ages, constrained by their relative stratigraphic position to one another (Richey et
al., 2020). The 10,000 modeled $CO_2$ estimates were trimmed by 28% to remove anomalously high/low values. The means of
the resulting 7,200 $CO_2$ estimates were compared to the trimmed means of the 10,000 $CO_2$ estimates to ensure that trimming
did not alter the central tendency of the data. Locality ages were resampled and perturbed assuming that the individual ages
and truncated age uncertainties represent the mean and standard deviation of the ages. Similarly, the trimmed $CO_2$/$O_2$:$CO_2$
datasets were resampled and the resampled ages and estimates were used to build 1000 resampled datasets. Each resampled
dataset was subjected to LOESS analysis (0.25 smoothing) and the median and 95% and 75% confidence intervals were
calculated (Figs. 2, 3a–b, S1). The Pennsylvanian and Permian portions of the record were analyzed separately due to
differing data density, with significant overlap across the Pennsylvanian-Permian boundary interval (Figs. 2b, 3b, S1b).

To test the validity of short-term fluctuations in the LOESS $CO_2$ trend, we undertook further analysis of the raw Monte

Carlo data produced by PBUQ and the mechanistic stomatal model in several short-term increments, by calculating
individual $CO_2$ data points via bootstrapping for each increment (Figs. 2b, S1b). Eleven short-term highs or lows (A–K on
Fig. 4A) were designated and used to form bins of ± 0.5 to ±1 Myr. Within an individual bin, each shown 'bootstrapped'
$CO_2$ data point is the trimmed mean of 10,000 Monte Carlo model runs. The Monte Carlo model runs for each data point
were sorted from lowest to highest $CO_2$ value and the lowest $CO_2$ values for each data point within the bin were averaged.
This averaging was repeated sequentially for each of the 10,000 values creating 10,000 means for each bin (n=11). To
evaluate whether a visually perceived rise or fall (e.g., A to B decrease or B and C increase) is statistically valid, the 10,000
means of two adjacent bins were compared sequentially with one another (i.e., the mean of the lowest value of one bin was
compared to the mean of the lowest value of the adjacent bin) in order to calculate a percent change $(((V_2 - V_1)/V_1) * 100)$
for each of the 10,000 model runs, resulting in 10,000 percent changes for each set of adjacent bins. The percent of the





10,000 comparisons that confirm an increase or decrease between bins is reported (Fig. 4B–J) as a measure of the statistical
significance of the short-term fluctuations in $CO_2$ concentration visually observed on the LOESS trend.

**3 Results**

Revised early Permian mineral-based $CO_2$ estimates define a substantially narrower range (45–1150 ppm; Fig. 2a) than

previous estimates (175–3500 ppm) made using the same pedogenic carbonate sample set (Montañez et al., 2007) while
maintaining the original trends and including fewer photosynthetically untenable concentrations ($\leq$170 ppm; Gerhart and
Ward, (2010)). New early Permian cuticle-based estimates show a high level of congruence by locality and broad plant
functional type, falling within the revised pedogenic-based $CO_2$ range (Figs. 2a, S1a). Similarly, stomatal-based estimates for
the four Pennsylvanian interglacial floras are within the estimated $pCO_2$ range defined by the pedogenic carbonates (Fig. 2a,
S1a) and late-glacial wetland plant fossils (Montañez et al., 2016). Notably, the newly integrated record confirms that
atmospheric $CO_2$ concentrations during Pennsylvanian interglacials ($10^4$-yr) were elevated (482 to 713 ppm [-28/+72 ppm])
relative to glacial periods (161 to 299 ppm [-96/+269 ppm]).

Overall, the new $pCO_2$ record documents declining $CO_2$ through the final 13-Myr of the Pennsylvanian into the earliest

Permian, including a 2.5-Myr interval (307 and 304.5 Ma) of minimum $CO_2$ values (<400 to ~200 ppm) in the Kasimovian
(Fig. 2b, S1b). Declining $pCO_2$ in the late Carboniferous coincides with rising atmospheric $pO_2$ (Glasspool et al., 2015;
Krause et al., 2018; Lenton et al., 2018); thus, $O_2$:$CO_2$ ratios in the interval of minimum Pennsylvanian $CO_2$ are nearly two
times those of present-day (~515; gray line in Fig. 3a). A 10-Myr CO2 nadir (~180 to < 400 ppm) characterizes the first two
stages (Asselian and Sakmarian; 298.9 to 290.1 Ma) of the early Permian, overlaps with the peak occurrence of glacial
deposits in the LPIA (gray boxes in Fig. 2b; Soreghan et al., (2019)), and defines a second interval of anomalously high
$O_2$:$CO_2$ ratios (up to 970 ppm; Fig. 3a). A subsequent long-term rise (~17 Myr) in $pCO_2$ to peak values up to ~740 ppm (-
190/+258 ppm) defines the remainder of the early Permian coincides with multiple episodes of extensive and long-lived
volcanism (Fig. 2b; Torsvik et al., (2008); Zhai et al., (2013); Sato et al. (2015); Shellnutt, (2018); Chen and Xu, (2019)).
This $pCO_2$ rise is also coincident with a decline in $O_2$:$CO_2$ to below present-day values (Fig. 2b, S1b, 3a).

Short-term intervals of rising or falling $CO_2$ in the LOESS trend, within dating uncertainties, coincide with a brief but



acute glaciation in the Kasimovian and with repeated deglaciations in south-central Gondwana in the early Permian (Griffis
et al., 2018; Griffis et al., 2019), as well as with restructuring of marine and terrestrial ecosystems (Figs. 3b-d). The statistical
significance of these short-term rises and falls in $CO_2$ was evaluated by analyzing the raw Monte Carlo estimates (10,000
model runs per data point shown on the LOESS trend) generated by the aforementioned $CO_2$ models (Breecker, 2013; Franks
et al., 2014), from which the bootstrapped $CO_2$ estimates for eleven increments of short-term rise or fall were subsequently
determined (Fig. 4a). The analysis of the Monte Carlo $CO_2$ estimates within these short-term intervals of rising or falling
$CO_2$ indicates that 72.5 to 100% of the data confirm a visually observed increasing or decreasing trend (Fig. 4).

**4 Discussion**
**4.1 Declining $CO_2$ through the Main Phase of the LPIA**
Atmospheric $CO_2$ concentrations in the final 13 Myr of the Carboniferous (the Pennsylvanian portion of our record) are
generally higher than those of the earliest Permian (Fig. 2b) and overall decline through the later part of the Carboniferous.
Higher $pCO_2$ in the latter half of the Pennsylvanian is compatible with the hypothesized waning of large Early to Middle
Pennsylvanian glaciers in the Late Pennsylvanian (c.f. Fielding et al., (2008), including widespread terminal deglaciation in a
major glacial depocenter in south-central Gondwana (Parana Basin, Brazil) toward the close of the Carboniferous (Griffis et
al., 2018; Griffis et al., 2019). Declining $pCO_2$ toward a nadir in the earliest Permian is also consistent with a renewed
increase in the geographic distribution of glacial deposits in Gondwana beginning in the Late Pennsylvanian and peaking
(apex) in the earliest Permian (Fig. 2b; Soreghan et al., (2019)).
A tectonically driven increase in $CO_2$ consumption via a strengthening of the silicate weathering ('climate stabilizing')
negative feedback (Walker et al., 1981; Berner and Caldeira, 1997) has been proposed as the driver of the Pennsylvanian
decline in $pCO_2$ (Goddéris et al., 2017). The strength of the negative feedback varies with the degree of 'weatherability' (i.e.,
the susceptibility to weathering), which, in turn, is predominantly controlled by the intensity of the hydrologic cycle
(precipitation and surface runoff), with further influence by surface temperature and vascular plants (Dessert et al., 2001;
Donnadieu et al., 2004; West, 2012; Maher and Chamberlain, 2014; Caves et al., 2016; Ibarra et al., 2016). Uplift of the
Central Pangaean Mountains (CPM) through the Pennsylvanian would have increased weatherability in the tropics by





inducing orographic precipitation and creating steeper slopes (Goddéris et al., 2017), thus providing a greater supply of fresh
mineral surfaces and enhanced surface runoff and fluid travel paths (cf. Maher and Chamberlain, 2014). Consequently,
CPM-induced increased weatherability and $CO_2$ consumption would have enhanced the global efficiency of weathering and
created a tighter coupling between $CO_2$ and climate at this time (cf. Maher and Chamberlain, (2014); Caves et al., (2016)).

The results of our GEOCLIM modeling, for a Himalayan-type mountain range (an analog for the CPM) and

parameterized such that 30% of the alkalinity generated by silicate weathering originates from the weathering of mafic rocks
(referred to as the 'reference continental silicate mineral assemblage or GEOCLIM_REG), indicates steady-state $CO_2$
concentrations (blue symbols and lines on Fig. 5A and B) that are well below the middle to late Carboniferous (340 to 300
Ma) threshold for initiation of continental ice sheets (840 ppm; Lowry et al., (2014). A hypothesized primary influence of the
CPM on $CO_2$ consumption through increased weatherability is further supported by the coincidence of modeled seawater and
marine proxy $^{87}Sr/^{86}Sr$ values that define a plateau of peak radiogenic values that is sustained for 15-Myr of the late
Carboniferous (318 to 303 Ma; Fig. 5b). The proxy-based seawater $^{87}Sr/^{86}Sr$ plateau has been long interpreted to record
exposure and weathering of uplifted and metamorphosed crustal rocks of the CPM that had radiogenic Sr isotope
compositions (Chen et al., (2018) and references within).

Additionally, the burial of substantial organic matter as peat in swamp environments prone to preservation (ultimately

as coal) during the Pennsylvanian would have partitioned global $CO_2$ consumption between silicate weathering and organic
carbon burial, further driving a lower steady-state $pCO_2$ (D'Antonio et al., 2019; Ibarra et al., 2019). Our modeling, however,
assumes a constant pre-Hercynian solid Earth degassing through the study interval and does not account for increased
magmatic $CO_2$ during Hercynian arc-continent collision and potential widespread eruptive volcanism in the late
Carboniferous (Soreghan et al., 2019), both of which could have increased steady-state $CO_2$.

Short-term fluctuations in $pCO_2$ superimposed on the 40-Myr record and confirmed as statistically significant (99.9 to

100% of estimates; Fig. 4b-d), coincide with major environmental and biotic events. The brief interval of minimum $pCO_2$ (an
average of ~300 ppm, but as low as 180 ppm) in the late Carboniferous (Kasimovian Stage, 307 to 304.5 Ma; Fig. 3b)
coincides with a short-lived but acute glaciation (306.5 to 305 Ma) recorded by prominent valley incision and large-scale
regression recorded by cyclothemic successions in the U.S. Appalachian Basin and Midcontinent, as well as the Donets



Basin, Ukraine (Belt et al., 2011; Eros et al., 2012; Montañez et al., 2016). Significant and repeated restructuring of wetland
forests throughout tropical Euramerica, involving quantitative changes in floral composition and dominance, occurred during
this 2.5 Myr $pCO_2$ minimum (and $O_2{:}CO_2$ maximum; Fig. 3a–c). Before the short-term $pCO_2$ low, Euramerican tropical
forests had expanded to their maximum aerial extent ($\geq 2 \times 10^6$ km$^2$) under $CO_2$ concentrations of ~500 ppm (Moscovian
Stage, Fig. 3b). The aerial extent of these forests dropped by half (green X in Fig. 3c; Cleal and Thomas, (2005)) coincident
with the decline in $pCO_2$ and near doubling of $O_2{:}CO_2$ (Fig. 3a–b). Moreover, within this $pCO_2$ low (Fig. 3b), arborescent
lycopsids of the wetland forests went extinct throughout Euramerica (white X in Fig. 3c) and seasonally dry tropical floras
shifted from cordaitalean- to walchian-dominance (~307–306.8 Ma; Fig. 3c; DiMichele et al., (2009); Falcon-Lang et al.,
(2018)). These restructuring events occurred at or proximal to $CO_2$ falling below 400 ppm, supporting a previously
hypothesized but untested $CO_2$ threshold for the Pennsylvanian ecologic turnovers (Fig. 3b–c; Beerling et al., (1998);
Beerling and Berner, (2000); Montañez et al., (2016). In the oceans, foraminiferal diversity decreased substantially during
the Kasimovian $pCO_2$ low with the loss of ~200 species (~58% of all taxa; 1[st] gray bar in Fig. 3d; Groves and Yue, (2009))
presumably due to decreasing seawater temperatures.

The interval of $CO_2$ minima was terminated by a rapid rise across the Kasimovian-Gzhelian boundary (303.7 Ma) to

$CO_2$ concentrations above 600 ppm (Fig. 2b; S1b). The short-term interval of elevated $pCO_2$ (304 to 302.5 Ma) is coincident
with a ~1.5‰ decline in seawater $\delta^{13}C$ (Grossman et al., 2008) compatible with a decline in the $CO_2$ sink provided by
terrestrial organic C (peats) burial (gray bar on Fig. 2b) and/or a peak in pyroclastic volcanism between ~310 and 301 Ma
(Soreghan et al., 2019). This period of increased $pCO_2$ overlaps with the Alykaevo Climatic Optimum (orange bar on Fig.
3c), defined by the invasion of tropical Euramerican vegetation into the *Ruflora*-dominated, mid-latitude Angaran floral
province (Cleal and Thomas, 2005). Terminal deglaciation in south-central Gondwana (Parana Basin, Brazil), U-Pb dated to
between ~302 and 298 Ma (Cagliari et al., 2016; Griffis et al., 2018), may have been linked to the Late Pennsylvanian
interval of elevated $CO_2$, although this requires further testing (Figs. 2b, 3b). Conversely to the Kasimovian $CO_2$ low, a
significant change in global diversity of foraminifera involving a doubling of species occurred during this subsequent period
of elevated $CO_2$ (black bar on Fig. 3d; Groves and Yue, (2009)).

**4.2 An Early Permian $CO_2$ Nadir**

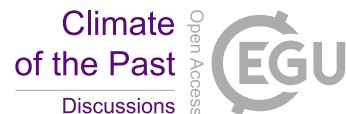

Atmospheric $p$CO$_2$ dropped substantially across the Carboniferous-Permian Boundary (i.e., 298.9 Ma) to a 10-Myr interval
(300–290 Ma) of the lowest concentrations (160 to <400 ppm) of the 40-Myr record (Fig. 2b). The CO$_2$ nadir, which spans
the Asselian and Sakmarian stages, coincides with renewed glaciation and maximum ice sheet extent, marking the apex of
LPIA glaciation (Fig. 2b; Fielding et al., (2008); Isbell et al., (2012); Montañez and Poulsen, (2013); Soreghan et al.,
(2019)), as well as with a large magnitude eustatic fall archived in paleotropical successions worldwide (Koch and Frank,
2011; Eros et al., 2012). Widespread glacial expansion temporally linked to this interval of lowest overall $p$CO$_2$ argues for
CO$_2$ as the primary driver of glaciation rather than recently proposed mechanisms, such as the influence of the closing of the
Precaspian Isthmus (Davydov, 2018) or a decrease in the radiative forcing resulting from increased atmospheric aerosols by
explosion volcanism at this time (Soreghan et al., 2019). The very low greenhouse radiative forcing associated with this low
CO$_2$ interval would have been amplified by 2.5% lower solar luminosity (Crowley and Baum, 1992), reduced transmission
of short-wave radiation (Poulsen et al., 2015) by the high $p$O$_2$ atmosphere of the early Permian (Krause et al., 2018; Lenton
et al., 2018), and by increased atmospheric aerosols at this time (Soreghan et al., 2019).
Notably, the 10-Myr $p$CO$_2$ nadir raises a paradox as to what was the primary CO$_2$ sink(s) at the time given that the CO$_2$
sinks of the Pennsylvanian were no longer prevalent. This paradox reflects the waning denudation rates of the CPM by the
early Permian (Goddéris et al., 2017), intensifying pantropical aridification, possibly driven by increasing continentality
(yellow to red bar in Fig. 3c; DiMichele et al., (2009); Tabor et al., (2013)), and the demise of the wetland tropical forests
and associated loss of peats before the close of the Carboniferous (black-to-gray bar in Fig. 2b; Hibbett et al., (2016)). In
turn, surface runoff would have been inhibited and the supply of fresh silicate minerals dampened, thus lowering overall
weatherability. Atmospheric CO$_2$ under the influence of these aforementioned environmental factors should have
equilibrated in the earliest Permian at a new higher steady-state level, even if solid Earth degassing did not increase (cf.
Gibbs et al., (1999)), thus raising a paradox.
The paradox, however, can be resolved if a switch in the ratio of mafic-to-granite rocks available for weathering
occurred with the turnover from the Carboniferous to the early Permian, in particular in the warm tropics. This reflects the
doubling or greater increase in weatherability of mafic mineral assemblages over granitic assemblages (Gaillardet et al.,
1999; Dessert et al., 2003; Ibarra et al., 2016), thus enhancing weathering efficiency and CO$_2$ drawdown, and creating a



tighter coupling between $CO_2$ and climate. In turn, the global silicate weathering flux needed to maintain homeostatic
balance in the carbon cycle for a given scenario can be maintained at a lower $pCO_2$ level.

Macdonald and others (2019) hypothesized that increased weatherability provided by the exhumation of ophiolites

along the ~10,000 km long Hercynian arc-continent suture zone, mainly situated in the paleotropics, was capable of lowering
$pCO_2$ below the ice initiation threshold in the Carboniferous, thus instigating the Late Paleozoic Ice Age. We used the
GEOCLIM model to interrogate this Carboniferous hypothesis further and to evaluate the potential of increased
weatherability, provided by increasing the ratio of outcropping mafic rocks to granite rocks available for weathering, as the
predominant driver of the early Permian $CO_2$ nadir. Figure 5 illustrates the influence of a successive increase in the surface
area of outcropping mafic rocks beginning with the reference continental silicate mineral assemblage (GEOCLIM-REG),
which was used to evaluate the influence of Pennsylvanian uplift of the CPM, to an up to 4-fold increase in the outcropping
of mafic rocks. In the GEOCLIM context, the weathering of mafic rocks is dependent on the surface of each grid cell, and of
the associated local runoff and air temperature, multiplied by a calibration constant. Increasing the exposure area of mafic
rocks is mathematically equivalent to multiplying the calibration constant.

Between 300 and 290 Ma, when predominant Pennsylvanian $CO_2$ sinks were lost (terrestrial organic C burial) or

waning (decreased precipitation and denudation rates of the CPM), modeled steady-state atmospheric $CO_2$ is maintained at
or below the $CO_2$ threshold for initiation of continental ice sheets (560 ppm; Lowry et al., (2014)) when the surface area of
outcropping mafic rocks is greater than 2-fold that of GEOCLIM-REG (Fig. 5a). Conversely, steady-state $CO_2$ rises well
above the glacial threshold (to 3500 pm) for the 'reference' continental silicate rock assemblage (Fig. 5a). Although
volcanism remained geographically extensive through the 10-Myr $CO_2$ nadir (Soreghan et al., 2019), the impact on
atmospheric $CO_2$ would have been short-lived ($\leq 10^5$ kyr; Lee and Dee, (2019)), and eclipsed on the longer term by the
increased weatherability provided by increased exposure of mafic rocks along the Hercynian arc-continent suture zone,
lowering steady-state $CO_2$ to potentially pre-volcanism levels (cf. Dessert et al., (2001)).

Independent evidence for a substantial shift in the partitioning of silicate weathering to more mafic mineral

assemblages in the earliest Permian exists in the late Paleozoic proxy-based seawater Sr isotope record, which documents a
rapid (0.000043/Myr) and near-linear decrease in seawater $^{87}Sr/^{86}Sr$ beginning in the latest Carboniferous (~303 Ma) and



continuing through into the middle Permian (Fig. 5b; Chen et al. (2018)). The simulated trends in seawater $^{87}Sr/^{86}Sr$ for
GEOCLIM-REG (blue line on Fig. 5b) through a 2- to 4-fold increase in the area of exposed mafic rocks capture the rapid
rise through the upper Carboniferous to peak values in the latter half of the Pennsylvanian and subsequent decline through
the early Permian. The rapid rate of decline in proxy $^{87}Sr/^{86}Sr$ values post-300 Ma, however, is best bracketed by simulated
$^{87}Sr/^{86}Sr$ for a 2- to 4-fold increase in mafic rock exposure. Moreover, the best fit of the simulated trends to the
geochronologically well-constrained bioapatite data (blue and green crosses on Fig. 5b) suggests a progressive increase in
mafic-to-granite ratio through the 10-Myr $CO_2$ nadir. This finding together with the need for minimally a 4-fold increase in
mafic rock outcropping in order to maintain $CO_2$ concentrations below the ice initiation threshold throughout the interval of
minimum $CO_2$ and apex of glaciation (Fig. 5), argues for a substantial increase in weatherability from the Carboniferous to
early Permian driven by a compositional shift in outcropping rocks available for weathering to a higher mafic-to-granite
ratio.

Although it has been suggested that peak ophiolite exhumation and maximum $CO_2$ consumption by their weathering

occurred in the late Carboniferous, thus initiating the LPIA (~330 to 300 Ma; Table S1 of Macdonald et al., (2019)), our
modeling results indicate that this is not compatible with proxy inferred moderate surface conditions of the late
Carboniferous (Montañez and Poulsen, 2013) and the radiation of forest ecosystems throughout the tropics (DiMichele,
2104). Increasing the surface area of outcropping mafic rocks (2- to 4-fold) during the Pennsylvanian results in steady-state
atmospheric $CO_2$ levels approaching Snowball Earth conditions given other operating influences on weatherability and $CO_2$
sequestration at the time (Fig. S6). For such conditions to be compatible with the paleontological record requires invoking a
substantial increase in solid Earth degassing rates. Alternatively, we hypothesize that the sustained $CO_2$ nadir and expansion
of ice sheets in the first 10 Myr of the Permian record a major reorganization of the predominant factors influencing
weatherability in the tropics across the Carboniferous-Permian transition, in particular, a substantial shift in the ratio of
mafic-to-granitic rocks available for weathering.

**4.3 Impact on Tropical Ecosystems**
The geologically rapid and large-magnitude drop in $pCO_2$ to a protracted minimum (Fig. 2b, S1b) and period of anomalously
high $O_2:CO_2$ (700 to 960; Fig. 3a) would have impacted earliest Permian terrestrial ecosystems given that modeling studies





indicate a decrease in photosynthetic rate and net primary productivity when plants are exposed to low (<400 ppm) $CO_2$
concentrations under elevated $pO_2$ (Beerling et al., 1998; Beerling and Berner, 2000). Euramerican tropical forests underwent
a permanent shift in plant dominance across the Carboniferous-Permian boundary interval from swamp-community floras to
seasonally dry vegetation (Black X on Fig. 3c), long attributed to intensification of an aridification trend that began in the
mid-Pennsylvanian (yellow to red bar in Fig. 3c; DiMichele et al., (2009); Tabor et al., (2013)). The high water-use
efficiency (WUE) of the seasonally dry plants would have made them water stress-tolerant. Furthermore, and analogous to
the vegetation turnover and extinction during the Pennsylvanian $CO_2$ minimum, the permanent shift in the tropics to
seasonally dry vegetation coincident with the earliest Permian drop in $pCO_2$ to below 400 ppm suggests a possible
ecophysiological advantage of these plants over the wetland floral dominants that they replaced  (Fig. 3a–c; c.f., Wilson et
al., (2017)). Moreover, this shift in vegetation dominance to plants of significantly higher WUE would have amplified the
aridification through a modeled ~50% decrease in canopy-scale transpiration (Wilson et al., 2017; Wilson et al., 2020). The
extreme habitat restriction of wetland floras was particularly consequential for tetrapods, leading to the acquisition of
terrestrial adaptions in crown tetrapods and the radiation and eventual dominance of dryland-adapted amniotes, possibly,
shaping the phylogeny of modern terrestrial vertebrates (Fig. 3c; Pardo et al., (2019)).

Notably, the $CO_2$ decline across the Carboniferous-Permian boundary into the 10-Myr nadir and early Permian peak in

$O_2$:$CO_2$ also corresponds to the evolution and radiation of glossopterids and gigantopterids (McLoughlin, 2011; Zhou et al.,
2017), with increasing vein density in the former (Fig. 3a–c; Srivastava, (1991)). These plant groups had complex,
angiosperm-like venation (Melville, 1983; Srivastava, 1991), with gigantopterids having the only known pre-Cretaceous
vessels in their stems, which would have increased their stem conductance of water (Li et al., 1996). Increased hydraulic
capacity provided by these morphological characteristics would have conferred a significant ecological advantage to these
plants under the low $CO_2$, high $O_2$, and elevated aridity conditions in which they evolved (cf. Gerhart and Ward, (2010); de
Boer et al., (2016)). In the oceans, a marked collapse in foraminiferal diversity with a notable fall in species to a minimum
from a Pennsylvanian zenith (425 to 110 species; Fig. 3d, e; Groves and Yue, (2009)) spanned the 10-Myr $pCO_2$ nadir,
analogous to the diversity drop during the Pennsylvanian low $CO_2$ interval.

Two statistically significant (94 to 100% on Fig. 4e–h), short-term increases in $pCO_2$ are superimposed on the early





Permian nadir (Fig. 3b ). The first (298 to 296 Ma) coincides, within age uncertainty, with a major deglaciation event in the
Karoo (southern Africa) and Kalahari (Namibia) basins of south-central Gondwana (296.41 Ma +0.27/-0.35 Ma; Griffis et
al., (2019)). The second short-term rise in $p$CO$_2$ (294.5 to 292.5 Ma) overlaps with the onset of widespread ice loss in several
southern Gondwanan ice centers (Fig. 2b; Soreghan et al., (2019)). This CO$_2$-deglaciation link suggests that continental ice
stability in the early Permian dropped substantially when $p$CO$_2$ rose above ~ 300 to 400 ppm and thus raises the question as
to whether the ice sheet CO$_2$ threshold was even lower than modeled (560 ppm; Lowry et al. 2014) during the earliest
Permian.

**4.4 CO$_2$-Forced Demise of the LPIA and Ecosystem Impact**
The 10-Myr CO$_2$ nadir terminated at 290 Ma with the onset of a protracted CO$_2$ rise that persisted to the highest levels of the
record (~740 ppm [-190/+258]) by the close of the early Permian (Fig. 2b). The onset of this protracted CO$_2$ rise overlaps
with initiation of a period of large-magnitude magmatism (red bars in Fig. 2b). Widespread volcanism began around 297.4
Ma (± 3.8 Ma) in northern Europe (Skagerrak-centered Large Igneous Province), extending well into Germany (Rotliegend)
(Torsvik et al., 2008; Käßner et al., 2019). The multi-stage Tarim magmatic episodes in China (292–272 Ma; with peaks at
~290 Ma and 280 Ma; Fig. 2b; Chen and Xu, (2019)), was likely associated with large magnitude CO$_2$ emissions given that
the magma, which distributed basalt (400 m thick) over a $2.5 \times 10^5$ km$^2$ region (Yang et al., 2013), intruded a thick
succession of early Paleozoic marine carbonates (Gao et al., 2017). The Panjal Traps, N.W. India (289 Ma ± 3 Ma;
(Shellnutt, 2018)) and the compositionally similar Qiangtang Dykes (283 Ma ± 2 Ma; Fig. 2b; Zhai et al., (2013)), albeit
relatively small in extent, were an additional potential volcanic CO$_2$ source, along with contemporaneous volcanism in
Oman. Furthermore, protracted Choiyoi volcanism, which began at 286.5 Ma ± 2.3 Ma (Sato et al., 2015) and continued over
~39 Myr in western Argentina, may have contributed substantial pulses of greenhouse gases in the early Permian (Spalletti
and Limarino, 2017). Once each magmatic episode waned, however, the mafic-dominated magmatic deposits would have
served as longer-term regional sinks leading to increased global CO$_2$ consumption (cf. Lee et al., (2015)). Thus, for steady-
state CO$_2$ to have increased through the remainder of the early Permian, the relative influence of CO$_2$ inputs must have
outpaced that of these, and other, outputs (CO$_2$ sinks).



Our modeled (GEOCLIM) steady-state $CO_2$ for a 4-fold increase in outcropping of mafic rocks surpasses the ice-sheet
initiation threshold at the termination of the $CO_2$ nadir (~290 Ma; red line and symbols on Fig. 5a), despite no change in
solid Earth degassing. That low $CO_2$ concentrations could no longer be maintained, despite a 4-fold increase in mafic rock
exposure, reflects overall intensifying aridification, denudation of the CPM, and a shift from dense forests to shrubby
savanna-like vegetation in Euramerica at this time. Thus, given that the magmatic $CO_2$ flux likely increased through the early
Permian, our model results indicate that maintaining low steady-state $CO_2$ concentrations during the $CO_2$ nadir would have
required an increasingly greater proportion of mafic rock weathering over the reference continental silicate mineral
assemblage of the Pennsylvanian, possibly well beyond a 4-fold increase.
A $CO_2$-forced demise of the Late Paleozoic ice age after 290 Ma is supported by the loss of continental ice from the
main ice depocenters in south-central Gondwana by 281.8 Ma ± 0.91 Ma (Griffis et al., 2018; 2019) and a 6-fold drop in
documented glacial deposits overall between the Sakmarian and Artinskian stages (Fig. 2b; Soreghan et al., 2019). The long-
term $CO_2$ rise through the remainder of the early Permian coincided with substantial marine and terrestrial ecosystem
perturbation (Fig. 3b–d; Chen and Xu, (2019)). In the marine biosphere, the uniformly low rates of global macroevolution in
marine organisms (brown bar on Fig. 3d) were reversed and broadly adapted and distributed genera reappeared, thus
restoring marine ecosystems to their pre-LPIA rates (Stanley and Powell, 2003). Pennsylvanian rugose corals (pink bar on
Fig. 3d) underwent a major turnover in composition to those that dominated until the End-Permian extinction and cold-
adapted marine bivalves and brachiopods turned over to warm-adapted forms across the Sakmarian-Artinskian boundary
(290.1 Ma; blue to red bar in Fig. 3d), synchronous with the onset of the long-term increase in $p$$CO_2$ (Wang et al., 2006);
Clapham and James, 2008). On land, the loss of pelycosaur families (three in the late Artinskian and four in the early
Kungurian (Kemp, 2006)) coincided with $CO_2$ sustained at >500 ppm. By the close of the Kungurian and the time of highest
$CO_2$ (740 ppm), basal synapsids largely disappeared and were replaced by more derived therapsids, tetrapod diversity
decreased significantly (Benton, 2012; McGhee, 2018), plant extinction rates reached a level comparable to that associated
with the extinction of arborescent lycopsids in the early Kasimovian (Cascales-Miñana et al., 2016), and
extinction/origination rates increased in fishes (Friedman and Sallan, 2012).





## 5 Conclusions

Glacial-interglacial climate cycles and large-scale glacioeustacy as well as repeated ecosystem change, analogous to that of the Pleistocene, characterized Earth's penultimate icehouse in the late Paleozoic. The dynamic glaciation history of this icehouse (the Late Paleozoic Ice Age (LPIA)) came to a close by the end of the early Permian with turnover to permanent greenhouse conditions. Thus, improved constraints on how atmospheric $p$CO$_2$ evolved during the LPIA and its subsequent demise is crucial for better understanding the role of greenhouse-gas forcing on Earth System processes during this time. The new and age-recalibrated $p$CO$_2$ reconstruction presented here for a 40-Myr interval (~313 to 273 Ma) of the late Paleozoic substantially refines existing Permian CO$_2$ estimates and provides perhaps the highest temporal resolution extended $p$CO$_2$ record prior to the Cenozoic. The multiproxy record confirms the previously hypothesized CO$_2$-glaciation linkage, including documenting the coincidence of a protracted period of minimum $p$CO$_2$ with inferred maximum ice extent during the earliest Permian. A long-term decline in $p$CO$_2$ through the late Carboniferous period of glaciation, culminating in the earliest Permian CO$_2$ nadir, lends support for a modeled progressive decrease in the CO$_2$ threshold for continental ice sheets through the LPIA.

Our new $p$CO$_2$ record provides the first stomatal-based evidence for elevated (up to 700 ppm) atmospheric CO$_2$ concentrations during short-term ($10^4$-yr) interglacials. Together with new O$_2$:CO$_2$ estimates of similar temporal resolution to $p$CO$_2$, the new atmospheric trends indicate a close temporal relationship to repeated ecosystem restructuring in the terrestrial and marine realms. In terrestrial ecosystems, the appearance and/or rise to dominance of plants with physiological and anatomical mechanisms for coping with CO$_2$ starvation and marked aridity correspond to drops in CO$_2$ below 400 ppm (as low as ~180 ppm) and O$_2$:CO$_2$ ratios nearly double those of late Paleozoic background values. Similarly, decreasing rates of macroevolution and diversity in the low-latitude oceans correspond to falling CO$_2$ to below 400 ppm. These CO$_2$–ecosystem relationships lead us to hypothesize that 400 ppm was an important threshold for ecosystem resilience during the late Paleozoic.

Modeling of steady-state $p$CO$_2$ during the late Paleozoic using an intermediate complexity climate-C cycle model (GEOCLIM) and comparison to the new multi-proxy CO$_2$ record provides new insight into the relative influences of the uplift of the Central Pangaean Mountains, intensifying aridification, and increasing mafic rock to-granite rock ratio of





outcropping rocks on the global efficiency of $CO_2$ consumption and secular change in steady-state $p$CO$_2$ through the late
Paleozoic. The simulations confirm that, for the Carboniferous and a continental silicate mineral assemblage for which 30%
of the alkalinity generated by silicate weathering originates from the weathering of mafic rocks, enhanced weatherability and
$CO_2$ consumption provided by the influence of the CPM on surface runoff and fresh mineral supply could have lowered
atmospheric $p$CO$_2$ well below the threshold for ice sheet initiation. Increasing the availability of mafic rocks for weathering
drives $CO_2$ levels toward snowball Earth conditions in the Carboniferous. Conversely, a substantial increase (up to 4-fold) in
the surface outcropping of mafic rocks over those modeled for the Carboniferous is needed to maintain the 10-Myr $CO_2$
nadir in the earliest Permian and is compatible with maximum exhumation of the Hercynian orogenic belt at this time as well
as with a rapid decline in seawater $^{87}$Sr/$^{86}$Sr inferred from biologic proxies. Although these findings support the hypothesis of
atmospheric $p$CO$_2$ response to uplift of the CPM as the primary driver for Carboniferous initiation of the LPIA (Goddéris et
al., 2017), they argue for a major reorganization of the predominant factors influencing weatherability in the tropics occurred
across the Carboniferous-Permian transition leading to decreased $p$CO$_2$ to values below 200 ppm. The demise of the LPIA
was greenhouse gas-forced reflecting the increasing importance of magmatic degassing and likely decreased weathering
efficiency driven by intensifying aridification, denudation of the CPM, and the loss of the wetland forests throughout tropical
Euramerica.
**Figures**



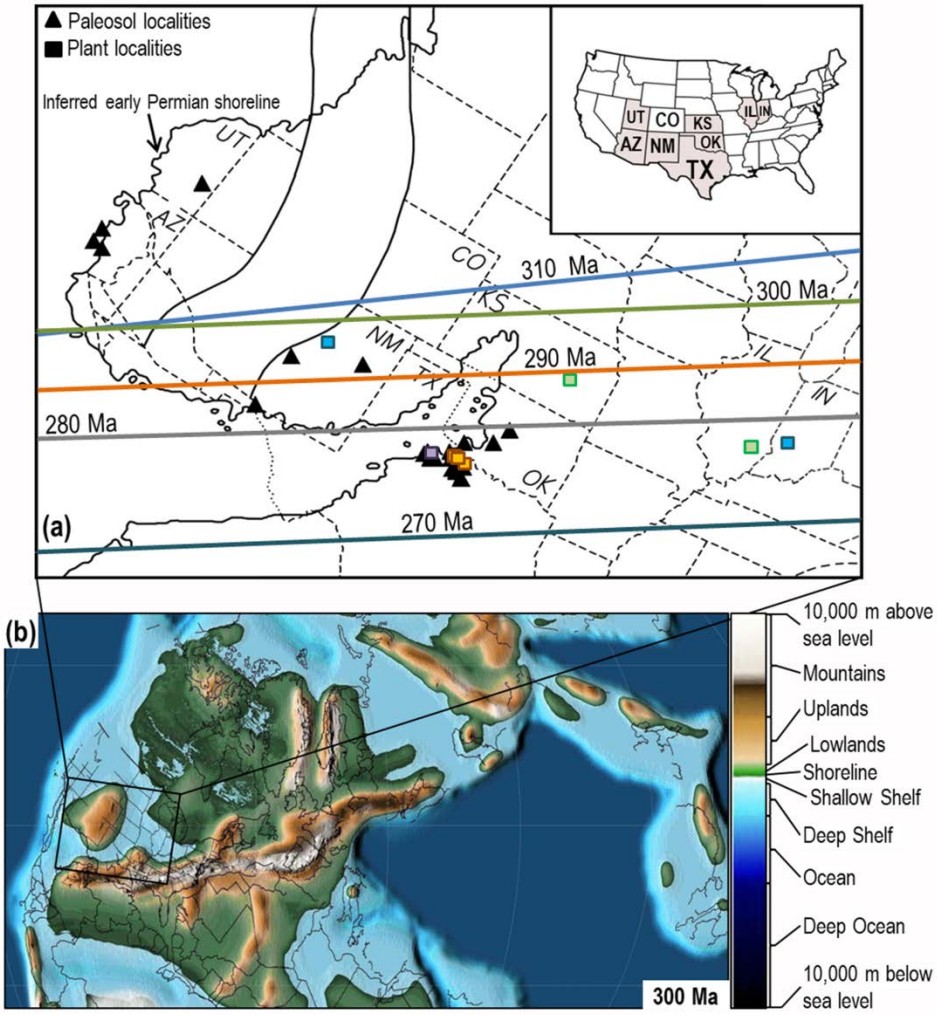

**Figure 1: Sampling localities in present-day and late Paleozoic geographic context**. **(a)** Sampling locations of pedogenic carbonates and plant fossils and their position relative to the Late Pennsylvanian (310 & 300 Ma) and early Permian (290 to 270 Ma) equator (the colors of the flora localities correspond to that of the paleo-equator at that time). White band traversing NM and CO is the area of inferred shortening during the Laramide and Sevier orogenies. Map modified from Montañez et al., (2007). **(b)** Earliest Permian (290 Ma) paleogeography (Scotese, 2016); shading corresponds to paleo-topographic/bathymetric scale on the right. Inset box is the location of panel (a).

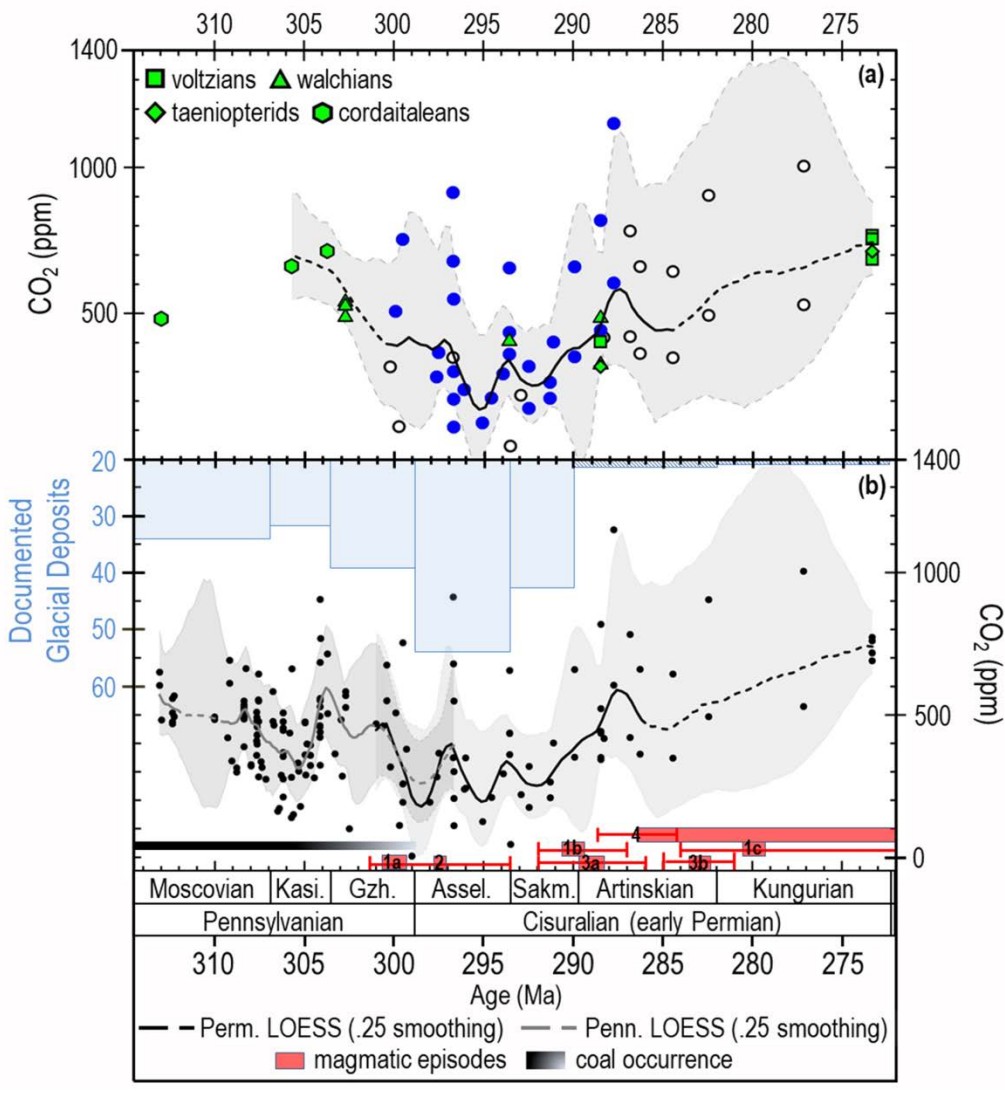

**Figure 2: Late Paleozoic CO$_2$ estimates. (a)** New and revised (Montañez et al., 2007) $p$CO$_2$ estimates, bootstrapped

LOESS trend, and 75% confidence interval (CI). Revised pedogenic carbonate-based estimates were made using $\delta^{13}C_{OOM}$

(blue filled circles; n = 28; Fig. S1) and $\delta^{13}C_{POM}$ (open black circles; n = 16; Fig. S1). Trendline is the median of 1000

bootstrapped LOESS analyses; dashed intervals indicate low data density and higher uncertainty. See Material and Methods

for details, Fig. S1 for error bars on individual CO$_2$ estimates and the 95% CI, and Richey et al. (2020) for the full dataset.

**(b)** Multiproxy CO$_2$ record and individual estimates (this study and age-recalibrated values of Montañez et al., (2016); n =

165), documented glacial deposits (Soreghan et al., 2019), and best estimate of timing (and uncertainties) of magmatic





episodes: 1a = Tarim 1, China (~300 Ma);  1b = Tarim 2 (292–287, peak ~290 Ma); 1c = Tarim 3 (284–272, peak ~ 280 Ma;
Chen and Xu, (2019)); 2 = Skagerrak-centered, NW Europe (297.5 ± 3.8 Ma; Torsvik et al., (2008)); 3a = Panjal Traps, NW
India (289 ± 3 Ma; Shellnutt, (2018)); 3b = Qiangtang Traps, Tibet (283 ± 2 Ma; Zhai et al., (2013)); 4 = Choiyoi, W
Argentina (beginning 286.5 Ma ± 2.3 Ma, continuing for up to 39 Myr; Sato et al., (2015)). Trendlines as in (A); dashed
intervals across the Carboniferous-Permian boundary (298.9 Ma) indicates overlap of the two LOESS trendlines.








**Figure 3: Late Paleozoic O$_2$:CO$_2$ and $p$CO$_2$, and comparison to environmental and biotic events. (a)** O$_2$:CO$_2$ estimates
using CO$_2$ values of this study and averaged time-equivalent modeled O$_2$ (Krause et al., 2018; Lenton et al., 2018). Trendline
is the median of 1000 bootstrapped LOESS analyses; gray horizontal line is present-day O$_2$:CO$_2$. **(b)** Bootstrapped
Pennsylvanian and Permian LOESS analyses (From Fig. 2A), with significant overlap across the Pennsylvanian- Permian
boundary interval, shaded to indicate CO$_2$ ranges. Temporal changes in terrestrial (c) and marine (d) ecosystems. Plant
biomes from Montañez (2016): Wetland Biome (WB) 1 (i.e., lycopsid-dominated), WB 2 (i.e., cordaitalean/lycopsid co-
dominance), WB 3 (i.e., tree fern-dominated), Dryland Biome (DB) 1 (i.e., cordaitalean-dominated), DB 2 (i.e., walchian-
dominated). Diagonal arrows indicate 10$^5$-yr glacial-interglacial shifts between wet- and dry-adapted floras.

**Figure 4: Analysis of statistical significance of short-term CO$_2$ fluctuations. (a)** White intervals (A—K) delineate short-term highs/lows in the CO$_2$ LOESS trend used for binning (n=11; bins ± 0.5 to 1 Myr resolution). Raw stomatal- and pedogenic carbonate-based CO$_2$ estimates generated by Monte Carlo analysis (10,000 model runs per CO$_2$ estimate; data in shaded intervals were not used). CO$_2$ between bins was compared by calculating the mean of the lowest through 10,000[th]





(highest) Monte Carlo values for all $CO_2$ points in each bin and comparing the means of the two bins sequentially. **(b)–(h)**
Histograms of the percent change between each of the 10,000 Monte Carlo means of the adjacent bins. Negative values
indicate a decrease in value between bins, positive values, an increase. The number above each histogram bar is of the
'percent change' values represented in each bar. The percent of the 10,000 model runs that confirm a given increase or
decrease in the LOESS trend is indicated by the % value shown on the right side of each panel. See Materials and Methods
for further details.











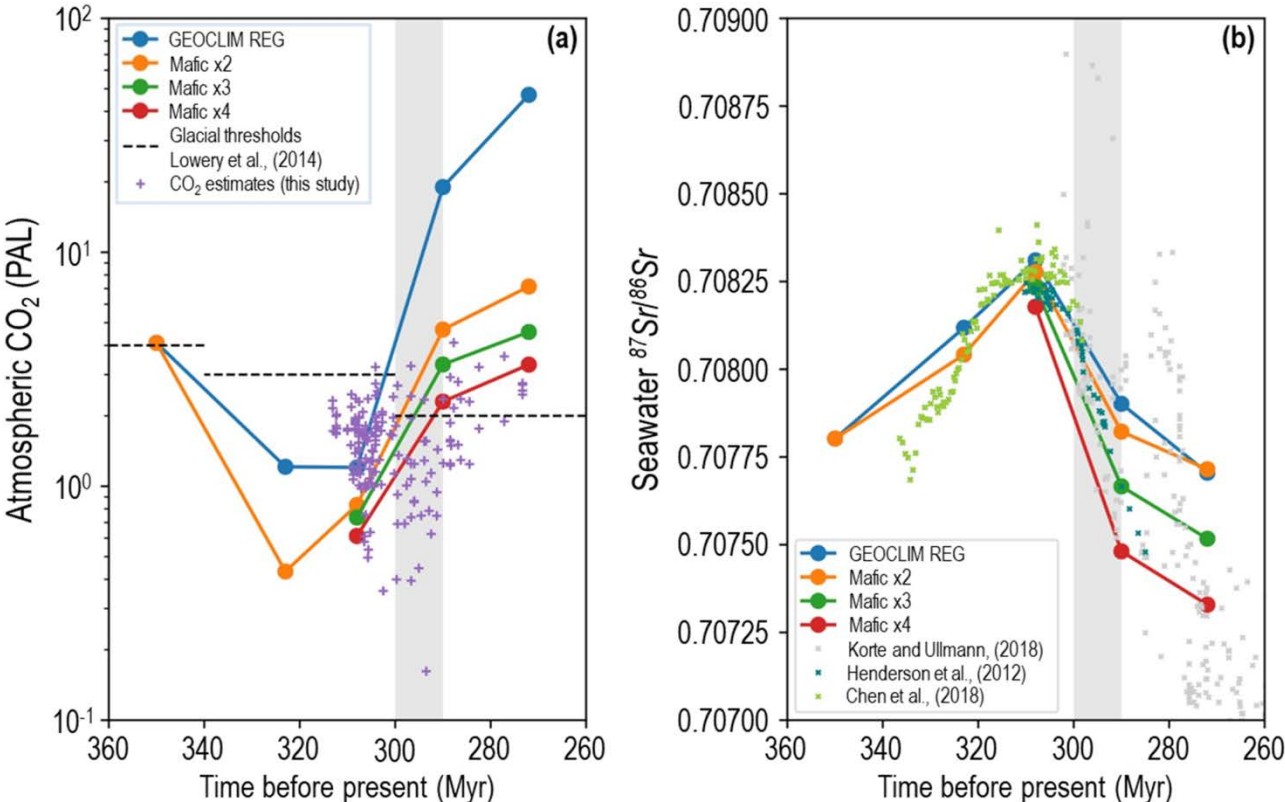


**Figure 5: Carboniferous through early Permian modeled (GEOCLIM) steady-state atmospheric CO$_2$ and seawater $^{87}$Sr/$^{86}$Sr for different surface areas of mafic rocks available for silicate weathering.** In the model, maximum geographic extent and altitude (5000 m) of the CPM is reached in the Moscovian (320 Ma), with altitude decreasing to 3000 m at 290 Ma and 2000 m at 270 Ma. **(a)** Simulated (color symbols and lines) and proxy $p$CO$_2$ estimates (purple crosses, this study). CO$_2$ thresholds for continental ice sheet initiation (360–340 Ma = 1120 ppm; 340–300 Ma = 840 ppm; 300–260 Ma = 560 ppm from Lowry et al., 2014) decrease in response to equatorward drift of Gondwana, favoring an overall reduction in ice-sheet size through time. The reference 'surface area of outcropping mafic rocks' (GEOCLIM REG) maintains steady-state atmospheric CO$_2$ below the ice initiation threshold from 350 to ~304 Ma. Steady-state atmospheric CO$_2$ for a 2-fold, 3-fold, and 4-fold increase in outcropping area of mafic rocks remains below the ice initiation threshold (560 ppm) up to ~300 Ma, crossing over at progressively later times in the early Permian. Threshold cross-over of steady-state CO$_2$ at ~290 Ma for a 4-fold increase in mafic rock exposure coincides with the termination of the 10-Myr CO$_2$ nadir (gray vertical bar; both panels).

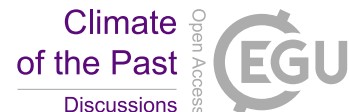

**(b)** Seawater $^{87}Sr/^{86}Sr$ modeled for the same set of varying surface areas of outcropping mafic rocks and $^{87}Sr/^{86}Sr$ values of
well-preserved biogenic calcites (gray filled squares) and conodont bioapatites (green and blue filled squares.

**Data Availability**
Underlying primary data is deposited in the Dryad Digital Repository (Richey et al., 2020) and can be accessed at
https://doi.org/10.25338/B8S90Q.

**Author contribution**
JDR and IPM designed the study. JDR collected the data, wrote the manuscript, and drafted the figures; IPM and YG carried
out the GEOCLIM modeling, wrote relevant parts of the manuscript, and drafted Fig. 5. All co-authors provided comments
on the manuscript.

**Competing interests**
The authors declare no competing financial interests.

**Funding**
This work was funded by NSF award EAR-1338281 to IPM and a National Science Foundation Graduate Research
Fellowship under University of California, Davis Grant #1148897 and a University of California, Davis Graduate Research
Mentorship Fellowship to JDR.

**Acknowledgments**
We thank C. Hotton (National Museum of Natural History Smithsonian Institute) and T. Taylor (R.I.P.), E. Taylor, and R.
Serbert (University of Kansas) for access to plant cuticle used in this study. We also thank B. Mills (University of Leeds) and
D. Temple-Lang and co-workers at the U.C. Davis Data Science Initiative for guidance with statistical analyses. Finally, we
thank J. White (Baylor University) for useful comments on the manuscript.



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
