# Peer review of "Influence of temporally varying weatherability on CO2-climate coupling and ecosystem change in the late Paleozoic."

_Climate of the Past, 2020_

## Referee Comment (RC1) · Anonymous Referee #1 · 10 Jun 2020

Richey J.D., et al: Influence of temporally varying weatherability on CO2-climate coupling and ecosystem change in the Late Paleozoic.

Journal: COP

Overview The question addressed in this manuscript concerns the carbon dioxide evolution at different time scale over the course of the Late Paleozoic (313-273Ma). Two pCO2 trends are presented and discussed, i) short-term intervals of rising/falling CO2 with a good time calibration and ii) a long trend (from 360 to 270Ma) driven by the weatherability of the Earth's surface. The first trend has been provided by a multi-proxy record offering a good temporal resolution while the long term reconstruction has been

obtained using a complex climate-carbon model exploring effects of tectonic and lithology (mafic rocks) and calibrated by the 87Sr/86Sr record. In addition to these points, the authors discuss interplays between Earth's climate (Late Paleozoic Ice Age) and ecosystem perturbations.

Scientific Interest Deep time climate and ecosystem reconstructions are challenging. Understanding how Earth's climate, tectonic and ecosystem modifications are linked represent an interesting advance. Consequently, this paper is an important contribution. Overall the article is well written however the discussion can be improved (not enough well organized). I identified several areas requiring clarification (listed below). These problems being easily solvable, I recommend a minor revision (ranked by order of importance).

Recommendation: Minor revision

(1) The discussion is not very clear. Indeed short-term variations and long-term processes are included in same sub-sections without to distinguish between modeling results and proxy (for instance lines 191-225 introduce modeling results while lines 226-243 present short-term pCO2 variations and biological turnovers. I do not think this presentation is very clear for the reader, indeed these parts have no links (or there is something lacking)). Moreover the discussion about ecosystem perturbations is interesting but has a modest impact to understand links between paleo-pCO2 and biological events. To highlight their results, the authors may consider to split their discussion (long-term vs short term) or creating a new sub-section for presenting modeling results.

(2) A few sentences of the discussion need to be rephrased or revised in order to reflect that initiation and deglaciation CO2-thresholds are different due to the climate hysteresis. Indeed the authors tend to consider the "CO2 glacial threshold" as an absolute value which determines the climate state of the Earth. The line 299 is correct because the final pCO2 (case at 270Ma, blue dote fig.5) is far above the glacial threshold however elsewhere even if the simulated CO2 overcomes the proposed glacial threshold,

that does not mean the termination of the Late Paleozoic Ice Age. ex : line 314 (the sentence can be removed) ex : line 383-390 (this issue can be solved by adding error bars for age determination for each steady state - indeed boundary conditions used to force climate models have their own uncertainties, especially paleomagnetic data used to reconstruct paleographies)

(3) fig.3b. the chosen colour are misleading and implicitly suggests "anomalies". Moreover authors seem to assume two climate states characterized by a threshold close to 400ppmv of CO2. This point needs more explanation (why this threshold is so different compared to values used in fig.5 and published by Lowry et al. 2014 ?)

(4) line 167. I don't understand how the duration of the "interglacial phase" has been estimated (104 yr). S6 suggests a range of values for the sedimentation rate. Why the duration does not seem to be affected by uncertainties (or explain why the duration does not depend on geological parameters)? In addition could you precise if the proposed duration (104 yr) is the mean value or the maximal value (or something else)? A brief paragraph summarizing limitations will be helpful for readers not familiar with this method.

---

## Referee Comment (RC2) · Anonymous Referee #2 · 12 Jun 2020

This paper improves the CO2 proxy record for the late Paleozoic and compares CO2 variations to other Earth system indices. Considerable care has been taken in assembling this record and evaluating it statistically, which is much appreciated and it will be a useful resource for the community. The paper also adapts previous modeling to assess what has driven the changes in CO2, and concludes that a change towards more reactive silicate lithology is necessary, for which there is independent support. Overall in my opinion it is a good, clear paper that needs little revision. I do have some minor revisions to suggest:

Line 39: typo "DiMichele, 2104"

Line 67: note the DOI address here does not currently work

Line 78: it is a bit confusing that this paper appears to cite itself? Again on line 137.

Line 112: Estimates of mean annual temperature are used to help determine past $CO_2$ levels. Any circularity should be considered here when going on to link the $CO_2$ estimates to climate.

Line 118: The Donnadieu paper cited is about the Cretaceous? Surely the model runs are not from that work?

Line 170: "307 and 304.5 Ma" should read "307 until 304.5 Ma"?, "<400 to ~200 ppm" also a bit confusing.

Line 173: missing subscript in CO2

Line 269: "Notably, the 10-Myr pCO2 nadir raises a paradox as to what was the primary $CO_2$ sink(s) at the time given that the $CO_2$ sinks of the Pennsylvanian were no longer prevalent. This paradox reflects the waning denudation rates of the CPM by the early Permian". Note that Joshi et al. (2019) in GRL have run climate model simulations for the earliest Permian and find higher silicate weathering rates as the denudation rate wanes. They argue that denudation rates are not a strong control on silicate weathering in mountains where the rate is high. Perhaps a weaker relationship between denudation and silicate weathering may help explain the paradox identified here?

Line 284: The comparison to Macdonald et al. is a little different in timing: their suture length reconstructions are small after 300 Ma.

Line 306: "rapid (0.000043/Myr)" use standard form here perhaps?

Line 319: "our modeling results indicate that this is not compatible with proxy inferred moderate surface conditions of the late Carboniferous" I would imagine many of the model parameters are not known well enough to really rule this out? Perhaps a more tentative statement here?

---

## Author Comment (AC1) · 24 Jun 2020

**Referee #1:**

*General Comment*:  "Deep time climate and ecosystem reconstructions are challenging. Understanding how Earth's climate, tectonic and ecosystem modifications are linked represent an interesting advance. Consequently, this paper is an important contribution. Overall the article is well written however the discussion can be improved (not enough well organized). I identified several areas requiring clarification (listed below). These problems being easily solvable, I recommend a minor revision (ranked by order of importance)."

*Response:* We thank reviewer #1 for their appreciation of this work.

*Comment (1)*: "The discussion is not very clear. Indeed short-term variations and long-term processes are included in same sub-sections without to distinguish between modeling results and proxy (for instance lines 191-225 introduce modeling results while lines 226-243 present short-term pCO2 variations and biological turnovers. I do not think this presentation is very clear for the reader, indeed these parts have no links (or there is something lacking)). Moreover the discussion about ecosystem perturbations is interesting but has a modest impact to understand links between paleo-pCO2 and biological events. To highlight their results, the authors may consider to split their discussion (long-term vs short term) or creating a new sub-section for presenting modeling results."

*Response:*  We, respectfully, do not agree that the discussion needs to be reorganized. We chose to present the discussion holistically by integrating modeling and proxy components via time increments. That is, we present three segments that not only correspond to three climatically and ecologically unique intervals (Middle to Late Pennsylvanian, Asselian and Sakmarian portion of the early Permian, and the remainder of the early Permian) but also correspond to long term $pCO_2$ trends and important superimposed short-term trends. We strongly feel that removing the short-term trends into a separate section results in loss of context in relation to the long-term trends throughout the record.

That said, in order to resolve reviewer #1's concern that short- and long-term term $CO_2$ variability and processes are presented together in the discussion, we have reorganized the manuscript in the following manner:

 We have altered lines 226-227 (**lines 230-232** after changes suggested by reviewers 1 and 2) to "Short-term fluctuations in $pCO_2$ are superimposed on the long-term decline through the latter portion of the Carboniferous. These short-term fluctuations have been confirmed as statistically significant (99.9 to 100% of estimates; Fig. 4b-d) and coincide with major environmental and biotic events." in order to provide a better segue the switch from discussion of the long-term trends to the superimposed short-term trends.

In addition, we have removed subsection 4.3 and rearranged and integrated that text into the latter portion of subsection 4.2 (**lines 351-382** after changes suggested by reviewers 1 and 2). In this manner, all sections in the discussion are now arranged by subsections that correspond to

time and $CO_2$ trends. Each subsection is structured such that the long-term proxy trends and model explanation of those long-term trends are discussed first, followed by discussion of short terms trend and their correlation ecosystem perturbation. This reconfiguration preserves the intended holistic presentation of the discussion while also clearing delineating long- and short-term trends within each subsection. We hope that this resolves the issue brought forth by reviewer 1.

***Comment (2):*** "A few sentences of the discussion need to be rephrased or revised in order to reflect that initiation and deglaciation CO2-thresholds are different due to the climate hysteresis. Indeed the authors tend to consider the "CO2 glacial threshold" as an absolute value which determines the climate state of the Earth. The line 299 is correct because the final pCO2 (case at 270Ma, blue dote fig.5) is far above the glacial threshold however elsewhere even if the simulated CO2 overcomes the proposed glacial threshold, that does not mean the termination of the Late Paleozoic Ice Age. ex : line 314 (the sentence can be removed) ex : line 383-390 (this issue can be solved by adding error bars for age determination for each steady state - indeed boundary conditions used to force climate models have their own uncertainties, especially paleomagnetic data used to reconstruct paleographies)"

***Response:*** We certainly did not intend to imply that the $CO_2$ threshold for initiation of continental ice was a threshold above which all glaciers would collapse. Also on the time scales at which we are dealing with in this paper (10s of thousands to millions of years), the time lag between the rise in $CO_2$ above a level at which continental glaciers can be sustained and the timing of glacier collapse determined by hysteresis (1000 of years) would not be discernable.

We have clarified the original statement (Line 314, **lines 335-339** after changes suggested by reviewers 1 and 2)) to address this by the following revision: "This finding, together with the hypothesized need (the aforementioned mechanism two) for minimally a 4-fold increase in mafic-rock outcropping in order to maintain $CO_2$ concentrations below the ice initiation threshold for a sustained period longer than that of hysteresis (i.e., throughout the interval of minimum $CO_2$ and apex of glaciation; Fig. 5), argues for a substantial increase in weatherability from the Carboniferous to early Permian driven by a compositional shift in outcropping rocks available for weathering to a higher mafic-to-granite ratio."

Concerning Lines 383-390 (**lines 401-409** after changes suggested by reviewers 1 and 2), we have added error bars to simulated steady-state $CO_2$ and $^{87}Sr/^{86}Sr$ trendlines, constrained by the simulated intervals (symbols on the figure) as requested.

***Comment (3):*** "fig.3b. the chosen colour are misleading and implicitly suggests "anomalies". Moreover authors seem to assume two climate states characterized by a threshold close to 400ppmv of CO2. This point needs more explanation (why this threshold is so different compared to values used in fig.5 and published by Lowry et al. 2014 ?)"

***Response:*** The 400 ppm value is not a threshold, but rather the mean value for the 16 millionyear record of atmospheric $p$CO$_2$ through the later Pennsylvanian reported in Montañez et al. 2016 and was used here as a guide solely. We have clarified this in the figure 3b caption (**lines 494-495** after changes suggested by reviewers 1 and 2).

*Comment (4)*: "line 167. I don't understand how the duration of the "interglacial phase" has been estimated (104 yr). S6 suggests a range of values for the sedimentation rate. Why the duration does not seem to be affected by uncertainties (or explain why the duration does not depend on geological parameters)? In addition could you precise if the proposed duration (104 yr) is the mean value or the maximal value (or something else)? A brief paragraph summarizing limitations will be helpful for readers not familiar with this method."

*Response:* The Midcontinent and Appalachian cyclothems from which many of the samples were obtained, are inferred as eccentricity cycles (Fielding et al. 2020). Fielding et al. 2020 has recently concluded that "geochronological constraints are consistent with each cycle representing a 100 ky (short eccentricity) interval, most likely related to waxing and waning of contemporaneous ice centers on Gondwana." In addition, given that interglacials of today have a duration of 10s of 1000s of years, by analogy, interglacials of the past are also 10s of 1000s of years in duration. We have revised Lines 166 to 168 (**lines 167-171** after changes suggested by reviewers 1 and 2) to clarify this. The sentence now reads: "Notably, the newly integrated record confirms elevated atmospheric CO$_2$ concentrations (482 to 713 ppm [-28/+72 ppm]) during Pennsylvanian interglacials in comparison to $p$CO$_2$ during glacial periods (161 to 299 ppm [-96/+269 ppm]), with interglacial durations on the order of 1000s to 10s of 1000s of years given the inferred eccentricity scale duration of the glacial-interglacial cycles (Horton et al. 2012; Montañez et al. 2016; Fielding et al. 2020)."

---

## Author Comment (AC2) · 24 Jun 2020

*General Comment*: "This paper improves the CO2 proxy record for the late Paleozoic and compares CO2 variations to other Earth system indices. Considerable care has been taken in assembling this record and evaluating it statistically, which is much appreciated and it will be a useful resource for the community. The paper also adapts previous modeling to assess what has driven the changes in CO2, and concludes that a change towards more reactive silicate lithology is necessary, for which there is independent support. Overall in my opinion it is a good, clear paper that needs little revision. I do have some minor revisions to suggest:"

*Response:* We thank reviewer #2 for their appreciation of this work and encouraging comments.

*Comment (1)*: "Line 39: typo "DiMichele, 2104"

*Response*: This has been fixed.

*Comment (2)*: "Line 67: note the DOI address here does not currently work"

*Response*: This was intentional. The underlying data has been deposited in the Dryad Digital Repository, but we chose to keep the data private during the process of peer review. If this work is accepted, we will make the data fully public. Until that time, the data set can be shared privately via a URL if requested by either the editor or reviewers.

*Comment (3)*: "Line 78: it is a bit confusing that this paper appears to cite itself? Again on line 137."

*Response:* That is not a citation of this paper, but the underlying data. The author guide to Climate of the Past mandates "the proper citation of data sets in the text and the reference list (see section references) including the persistent identifier." We have cited the dataset as Richey et al. 2020 to comply with these instructions. However, we have altered all of the in-text citations of the dataset to include the DOI and make it clear that the dataset is being cited (**lines 74, 79, 82, 138, 479** after changes suggested by reviewers 1 and 2). If we have misunderstood the instructions on how and when to cite the underlying data, please let us know and we will make any necessary changes.

*Comment (4)*: "Line 112: Estimates of mean annual temperature are used to help determine past CO2 levels. Any circularity should be considered here when going on to link the CO2 estimates to climate."

*Response:* Yes, this is correct; we used mean annual air temperatures as input for the PBUQ model to estimate the paleo-$CO_2$ estimates in cases where the paleosol estimates were reformulated in this study. For the part of the paleosol-based reconstruction that comes from Montañez et al. 2016 (i.e., the Pennsylvanian and earliest Permian estimates), a broad range of temperatures of 20 to 26°C (i.e., 23°C ±3°) was prescribed. For the estimates from Montañez et al. 2007 reformulated in this study (most Permian paleosol estimates), we use proxy soil

temperatures that come from many of the same paleosols (Tabor and Montañez 2005; Tabor et al., 2013). For the latter, for intervals with proxy soil temperatures of > 30°C, we used temperatures 5°C lower as the MAAT, for proxy soil temperatures of >25°C to ≤ 30°C, we used temperatures 3°C lower, and for temperatures ≤ 25°C, we used the actual proxy value. This scheme resulted in MAAT temperatures that range from 23 to 30°C. The error on these temperatures was assigned at ±3°C, like the estimates from Montañez et al. 2016. Despite the differences in the method by which MAAT was prescribed or calculated, out of the 103 paleosol-based estimates, only 5 MAAT values used fall out of the range of 20 to 26°C (i.e., 23°C ±3°).

These MAAT estimates are purposefully broad, given the uncertainty in paleo-temperatures for these past periods. However, the temperature ranges overlap with the range (18 to 26°C) indicated by the climate modeling for the terrestrial realm of the Pennsylvanian and early Permian Pangaean tropics (Poulsen et al. 2007; Montañez and Poulsen, 2013; Macarewich et al. in revision (*EPSL*)). In addition, the temperatures used overlap with the lower range of the pedogenic phyllosilicate temperatures (23 to 32°C) published by Rosenau and Tabor (2013). Importantly, there is no circular reasoning involved in using these values, as the reviewer raised as a concern, as these temperature estimates of 20 to 26°C encapsulate the minimum and maximum temperatures simulated by a GCM (GENESIS3; Horton et al. 2010; 2012) and an Earth System Model (iCESM 1.2; Macarewich et al. 2019; in revision) for the continental tropics over a $CO_2$ range of 280 to 840 ppm (overlapping the range of $CO_2$ calculated in the LOESS analysis in this study (175 to 750 ppm). Thus, by using the full range of MAATS (20 to 26°C, rarely >26°C) consistently throughout the modeling of the samples of Pennsylvanian and earliest Permian age, we feel we have conservatively represented the realistic MAATs in the paleotropics during the late Paleozoic in a manner that precludes circularity.

*Comment (5)*: "Line 118: The Donnadieu paper cited is about the Cretaceous? Surely the model runs are not from that work?"

*Response:* The Donnadieu et al. 2016 paper was solely referenced for the model and methods – not the results. However, after review, we have decided that Donnadieu et al. (2006) would be a more appropriate citation for model and methods than Donnadieu et al. (2016). We have addressed this removing Donnadieu et al. (2016) and adding "and approach as described in Donnadieu et al. (2006)." to the statement. The revised sentence now reads "The spatial distributions of the mean annual runoff and surface temperature were calculated offline for five time increments (Goddéris et al., 2017) covering the period of interest and for various atmospheric $CO_2$ levels using the 3D ocean-atmosphere climate model FOAM and the approach as described in Donnadieu et al., (2006) (**lines 117-119** after changes suggested by reviewers 1 and 2).

*Comment (6)*: Line 170: "307 and 304.5 Ma" should read "307 until 304.5 Ma"?, "<400 to ←⋯200 ppm" also a bit confusing.

*Response:* This has been changed to "…2.5-Myr interval (307 to 304.5 Ma) of minimum $CO_2$ values (less than 400 to as low as 200 ppm)…" (**line 173** after changes suggested by reviewers 1 and 2).

*Comment (7)*: Line 173: missing subscript in CO2

*Response:* This has been fixed (**line 176** after changes suggested by reviewers 1 and 2).

*Comment (8)*: Line 269: "Notably, the 10-Myr pCO2 nadir raises a paradox as to what was the primary CO2 sink(s) at the time given that the CO2 sinks of the Pennsylvanian were no longer prevalent. This paradox reflects the waning denudation rates of the CPM by the early Permian". Note that Joshi et al. (2019) in GRL have run climate model simulations for the earliest Permian and find higher silicate weathering rates as the denudation rate wanes. They argue that denudation rates are not a strong control on silicate weathering in mountains where the rate is high. Perhaps a weaker relationship between denudation and silicate weathering may help explain the paradox identified here?

*Response:* We thank reviewer 1 for bringing to our attention this very important paper. Indeed, Joshi et al.'s (2019) modeling results would support the idea of a delayed capacitor-discharge mechanism as the origin of the long-term decline in $p$CO$_2$ through the last 16 Myr of the Carboniferous (in our record) from ~500 ppm to <300 ppm by the earliest Permian, as well as the return to rising $p$CO$_2$ (to >500 ppm) after 10 million years into the early Permian.

However, we think that we must delve deeper into the respective models. The main improvement of Joshi et al. (2019) compared to GEOCLIM is higher spatial resolution. The model used by Joshi et al. (2019) allows a better representation of runoff, and hence, weathering, especially in the Central Pangean Mountains (CPM). However, the major difference between both models is the absence of climate dependence in the calculation of the spatially resolved physical erosion in Joshi et al. (2019). In their model, physical erosion is only dependent on the prescribed altitude of each grid cell, meaning that physical erosion is an external forcing of the model. This has major implications for the results of the Joshi et al. (2019) model. Indeed, when the CPM are high, the drop in temperature limits weathering rates, without compensation by enhanced runoff linked to orographic impact on the atmospheric circulation. Consequently, in the Joshi et al. (2019) model, the maximum weathering is reached when the mountains are already eroded (due to temperature rise at lower altitude), but physical erosion is also a function of runoff. In GEOCLIM, the dependence of the physical erosion on runoff does not allow the existence of such a delay between the maximum altitude of the CPM and the lowest atmospheric CO2.

Thus, we have a new paragraph to address this in Section 4.2 An Early Permian CO$_2$ Nadir (see **lines 284-304** after changes suggested by reviewers 1 and 2). We hope that this change provides a balanced discussion of our and Joshi et al., (2019) work. The new paragraph reads:

"Two mechanisms have the potential to resolve this paradox. The first, referred to as a delayed climate-controlled capacitor (Joshi et al. 2019), leads to a multi-million-year delay between the timing of peak orogenic uplift and maximum chemical weathering potential and CO$_2$ drawdown due to substantial differences in chemical weathering rates during the different phases of an orogenic cycle. In their study, the highest intensity of chemical weathering and capacity for CO$_2$ consumption occurs when mountains have been somewhat denuded rather than during peak uplift, reflecting the disproportionate influence of runoff temperature over hydrology and erosion on weathering potential. Notably, Joshi et al.'s (2019) coupled climate and geochemical modeling of the Late Paleozoic Ice Age yield an evolution of simulated $p$CO$_2$ over the period of

uplift and denudation of the CPM that corresponds both in absolute $CO_2$ concentrations and magnitude of change over this period (~320 to 290 Ma). That said, in Joshi et al. (2019), the physical erosion parameter is not dependent on climate, but, rather, is defined by the prescribed altitude. Thus physical erosion is an external forcing in their model. The absence of runoff dependence for physical erosion (as is the case for GEOCLIM) and the strong dependence of weathering on temperature may be the trigger for their simulated delay between maximum uplift and the highest intensity of $CO_2$ consumption by silicate weathering. In GEOCLIM, the dependence of the physical erosion on runoff does not allow for a millions of years delay between maximum uplift of the CPM and lowest simulated $pCO_2$. Further study is needed to interrogate the influence of this approach on the results.

The second mechanism, proposed here, is a substantial shift in the ratio of mafic-to-granite rocks available for weathering from the latest Carboniferous to the early Permian. This reflects the doubling or greater increase in weatherability of mafic mineral assemblages over granitic assemblages (Gaillardet et al., 1999; Dessert et al., 2003; Ibarra et al., 2016), thus enhancing weathering efficiency and $CO_2$ drawdown, and creating a tighter coupling between $CO_2$ and climate. In turn, with tighter coupling between $CO_2$ and climate, the global silicate weathering flux needed to maintain homeostatic balance in the carbon cycle for a given scenario can be attained at a lower $pCO_2$ level."

*Comment (9)*: Line 284: The comparison to Macdonald et al. is a little different in timing: their suture length reconstructions are small after 300 Ma.

*Response:* We are not sure whether we have misunderstood this comment. We agree that the compilation of suture zones made by Macdonald et al. (2019) indicates that the ~10,000 km long Hercynian arc-continent suture zone (in the paleotropics) was at a peak prior to 300 Ma (transition from Carboniferous to Permian). Lines 286–289 (**lines 308-310** after changes suggested by reviewers 1 and 2), state that we used GEOCLIM to test the Macdonald et al. 2019 hypothesis that the influence of increased mafic (ophiolites in their study) on $pCO_2$ was greatest in the ***Carboniferous***. As well as to "to evaluate the potential of increased weatherability, provided by increasing the ratio of outcropping mafic rocks to granite rocks available for weathering, as the predominant driver of ***the early Permian*** $CO_2$ nadir. We may have confused the reader by referring to both goals of the modeling in this section.

This has been resolved by revising the sentence (**lines 308-310** after changes suggested by reviewers 1 and 2) to read "Here, we used the GEOCLIM model to, first, interrogate this Carboniferous hypothesis further and, second, to evaluate the potential of increased weatherability, provided by increasing the ratio of outcropping mafic rocks to granite rocks available for weathering, as the predominant driver of the early Permian $CO_2$ nadir."

*Comment (10)*: Line 306: "rapid (0.000043/Myr)" use standard form here perhaps?

*Response:* This has not been changed as we believe the reviewer was asking that we change Myr to Ma. However, this would be incorrect as Ma is for a specific time vs. Myr for an increment of time (here 1 million years).

***Comment (11):*** Line 319: "our modeling results indicate that this is not compatible with proxy inferred moderate surface conditions of the late Carboniferous" I would imagine many of the model parameters are not known well enough to really rule this out? Perhaps a more tentative statement here?

***Response:*** We very much appreciate the reviewer's comment and we agree that the model parameters are associated with uncertainty. The results that we refer to in this section of the Discussion (Lines 284-328, **lines 305-339** after changes suggested by reviewers 1 and 2) are $1^{st}$-order differences in steady-state $p$CO$_2$ that would lead to climate regimes, which differ remarkably from one another. For example, the modeled steady-state $p$CO$_2$ for a 2- to 4-fold increase in the surface outcropping of mafic rocks available for weathering in the Pennsylvanian leads to near Snowball Earth conditions, which are incompatible with other earth system conditions at that time (from the literature). Conversely, modeling with the reference continental silicate mineral assemblage (GEOCLIM-REG) maintains the steady-state $p$CO$_2$ below the threshold for initiation of continental ice sheets but above unreasonably low values (<200 ppm). However, as the reviewer points out, there are uncertainties in the modeling. For example, if the solid Earth degassing rates increased through the Pennsylvanian (we invoke a constant CO$_2$ degassing rate), then it is feasible that an increased component of weathering of mafic rocks would have maintained sufficiently high CO$_2$ concentrations to accommodate the independent evidence for surface conditions at this time.

To that end, we have tempered the statements in Lines 318 to 328 (**lines 340-350** after changes suggested by reviewers 1 and 2) by revising the text as follows:

"If peak ophiolite exhumation and maximum CO$_2$ consumption by their weathering occurred in the late Carboniferous, thus initiating the LPIA (~330 to 300 Ma) as has been suggested (Table S1 of Macdonald et al., 2019), then our modeling results suggest that a substantial increase in solid Earth degassing rate at this time would have been necessary. In our simulation, increasing the surface area of outcropping mafic rocks (2- to 4-fold) during the Pennsylvanian results in steady-state atmospheric CO$_2$ levels approaching Snowball Earth conditions given other operating influences on weatherability and CO$_2$ sequestration at the time and no change in degassing rate (Fig. S6). Such conditions are not compatible with proxy inferred moderate surface conditions of the late Carboniferous (Montañez and Poulsen, 2013) and the radiation of forest ecosystems throughout the tropics (DiMichele, 2014). Rather, we hypothesize that the sustained CO$_2$ nadir and expansion of ice sheets in the first 10 Myr of the Permian record a major reorganization of the predominant factors influencing weatherability in the tropics across the Carboniferous-Permian transition, in particular, a substantial shift in the ratio of mafic-to-granitic rocks available for weathering."